# Natural Hollow Fiber-Derived Carbon Microtube with Broadband Microwave Attenuation Capacity

**DOI:** 10.3390/polym14214501

**Published:** 2022-10-24

**Authors:** Yanfang Zhao, Aichun Long, Pengfei Zhao, Lusheng Liao, Rui Wang, Gaorong Li, Bingbing Wang, Xiaoxue Liao, Rentong Yu, Jianhe Liao

**Affiliations:** 1Key Laboratory of Advanced Materials of Tropical Island Resources of Ministry of Education, School of Materials Science and Engineering, Hainan University, Haikou 570228, China; 2Guangdong Provincial Key Laboratory of Natural Rubber Processing, Agricultural Products Processing Research Institute, Chinese Academy of Tropical Agricultural Sciences, Zhanjiang 524001, China; 3Hainan Provincial Key Laboratory of Natural Rubber Processing, Zhanjiang 524001, China

**Keywords:** biomass-derived carbon microtube, kapok fiber, electromagnetic loss, microwave attenuation

## Abstract

Constructing hierarchical structures is indispensable to tuning the electromagnetic properties of carbon-based materials. Here, carbon microtubes with nanometer wall thickness and micrometer diameter were fabricated by a feasible approach with economical and sustainable kapok fiber. The carbonized kapok fiber (CKF) exhibits microscale pores from the inherent porous templates as well as pyrolysis-induced nanopores inside the wall, affording the hierarchical carbon microtube with excellent microwave absorbing performance over broad frequency. Particularly, CKF-650 exhibits an optimized reflection loss (RL) of −62.46 dB (10.32 GHz, 2.2 mm), while CKF-600 demonstrates an effective absorption bandwidth (RL < −10 dB) of 6.80 GHz (11.20–18.00 GHz, 2.8 mm). Moreover, more than 90% of the incident electromagnetic wave ranging from 2.88 GHz to 18.00 GHz can be dissipated by simply controlling the carbonization temperature of KF and/or the thickness of the carbon-microtube-based absorber. These encouraging findings provide a facile alternative route to fabricate microwave absorbers with broadband attenuation capacity by utilizing sustainable biomass.

## 1. Introduction

As one of the driving forces of the evolution of society, electronic information technology has facilitated the development of national defense security and civil daily life. At the same time, the surge of high-tech military equipment and ever-changing digital devices inevitably generate increasingly severe electromagnetic pollution, interrupting the normal running of equipment and posing a threat to the health of organisms. Therefore, developing various advanced absorbers for microwave attenuation is essential, especially those with light weight, high efficiency, and wide absorption frequency ranges [1]. In general, these absorbers can be categorized into magnetic loss materials, dielectric loss materials, and their combination [2]. Typical magnetic loss-dominated absorbers are mainly magnetic metals and ferrites, which is not an ideal choice due to their inherent drawbacks including bulkiness, inferior chemical resistance, and excess loading. Alternatively, dielectric loss microwave absorbing materials are regarded as promising candidates owing to their low density, robust physiochemical stability, and tailorable electromagnetic characteristics, which have captured the interest of industry and academia.

Over the past few decades, extensive efforts dedicated to fabricating carbonaceous microwave absorbers (such as carbon nanotube [3], graphene [4], carbon fibers [5], metal-organic framework [6], etc.) have demonstrated that constructing a tailor-made architecture is a versatile approach to tailoring the electromagnetic characteristics of microwave absorbers, affording excellent microwave attenuation (MA) capacity. Foams, sponges, and aerogels are typical three-dimensional (3D) structures integrating continuous macro holes with numerous internal nanopores [7], where macro holes serve as dihedral angles to multiply scatter/reflect penetrated microwave and nanopores act as an “effective medium” to improve impedance matching. In light of this concept, an increasing number of hierarchically porous carbonaceous materials for microwave attenuation have been developed by adopting many strategies including non-template and template methods [8,9,10]. For instance, hollow microspheres with mesopores constructed by a sacrificial templating show a minimum reflection loss (RL_min_) of −84.00 dB and an effective absorption bandwidth of 4.8 GHz, which are superior to those having only hollow microspheres or those with solid carbon microspheres [11]. Such mesoporous carbon hollow microspheres not only can efficiently attenuate electromagnetic energy via interior cavity-induced multiple reflection and scattering, but also achieve well-matched impedance through abundant solid-air effective media. Moreover, by freeze-drying and hydrazine hydrate vapor reduction methods, Li et al. fabricated metamaterials graphene sponge units with sphere, cubic, hexagonal prism, and frustum pyramid structures, demonstrating excellent impedance matching and then an ultrabroad EAB [12]. Nevertheless, these strategies have been constrained due to expensive precursors and environment-hazardous synthesis. Therefore, it is highly desirable to explore sustainable and economic raw materials accompanied by a facile fabrication method.

Owing to their renewable, eco-friendly, and abundant resources, biological-organisms-derived porous carbon materials have been extensively investigated and widely used in energy storage, catalysts, and pollutant adsorbents [13,14,15]. Similar to other man-made porous materials, it is promising to use bio-inspired microstructure to fabricate microwave absorbers because long-term evolution endows biomass with many unique features including hierarchical texture, periodic pattern, and distinct nanoarchitecture [16,17,18]. Inspired by this, many strategies have been adopted to improve the electromagnetic properties of porous carbon-based microwave absorbers, including optimizing the size of pores, enlarging surface areas, and doping with other elements [19]. Chen et al. fabricated chitosan-derived carbon aerogels with bamboo tube structure by controlling the growth speed/direction of ice crystals and frozen drying, demonstrating an RL_min_ of −46.30 dB and an EAB of 3.72 GHz [20]. Moreover, Qiu et.al confirmed that the MA capacity of walnut-shell-derived porous carbon activated by potassium hydroxide is much better than that of its only carbonized counterparts [21]. Additionally, three different wheat-flour-derived nanoporous carbon media can be obtained by controlling the activation time, where one with a 3D interconnected skeleton demonstrates an RL_min_ of −51 dB, which is much higher than that of those with honeycomb-like structures or irregular lump shapes [22]. Interestingly, nitrogen-doped cotton carbon monolith shows two times the MA capacity when it is compared with a solely carbonized one [19]. Although the introduction of magnetic metals into biomass-derived porous carbon can boost its MA capacity [11,23,24], these processes still involve harmful reagents, high-density ingredients, and complex synthesis. Consequently, it is of great importance and a challenge to design high-performance microwave absorbers directly from biomass-derived porous carbon.

Compared with random porous carbon MA materials that may result in excessive reflection at the interface, tubular structure with regular pores is of significance in addressing electromagnetic pollution [25]. Various strategies, including sacrifice templating [26] and chemical vapor deposition [27], are adopted to fabricate microtubes. However, most of these reported approaches are man-made, which increases the cost and environmental concern for large-scale fabrication of MA materials. As a kind of renewable resource, tubular morphologies can be found in wood, catkin, polar bear hair, and cancellous bone [28], which has been confirmed to be a good candidate for microwave attenuation [29]. Herein, a carbon-microtube-based sustainable microwave absorber was prepared via a facile one-step carbonization process by using kapok fiber (KF) as a precursor, and the influences of temperature on the nanostructure, morphology, and MA capacity of the as-fabricated samples are systematically investigated. Temperature-tailored pyrolysis affords the carbonized kapok fibers (CKF) with hierarchical structures, demonstrating the optimized MA capacity with an RL_min_ value of −62.46 dB (2.2 mm, 10.32 GHz) at 650 °C and a broad EAB of 6.80 GHz (2.8 mm, 10.4–18 GHz) at 600 °C. Morphology-associated electromagnetic behavior analysis reveals that such remarkable MA capacity resulted from hierarchical-structure-improved impedance matching as well as multiple attenuations originating from continuous carbon skeleton macro holes, interior nanopores, and defects/oxygen.

## 2. Materials and Methods

### 2.1. Materials

KF was harvested from the South Subtropical Botanical Garden in the Chinese Academy of Tropical Agricultural Sciences (Zhanjiang, China). High-purity argon (>99.999%) was provided by the Zhanjiang Oxygen Factory (Zhanjiang, China). Absolute ethyl alcohol was purchased from Aladdin Biochemical Technology Co., Ltd. (Shanghai, China). All of the chemical reagents in the experiment were analytically pure and used without further purification.

### 2.2. Preparation of KF-Derived Carbon Microtube

Carbon microtube was prepared via a feasible approach from economical and sustainable kapok fiber (KF), which is schematically illustrated in Figure 1. Typically, the collected KF was washed with distilled water to remove impurities and vacuum dried at 60 °C for 12 h. Then, the as-dried KF was placed into a ceramic crucible and transferred into a furnace tube. After being bowed with argon for 30 min to drive the remaining air in the quartz, the furnace tube was heated via a programmed temperature from room temperature to 500 °C, 550 °C, 600 °C, 650 °C, 700 °C, 750 °C, or 800 °C with a heating rate of 10 °C/min, respectively. After being kept at the given temperature for 2 h and then cooled down to room temperature, a fluffy black product was obtained and designated as CKF-x, where x refers to the carbonization temperature.

### 2.3. Characterizations

**XRD.** Phase constitution of the sample was investigated with a D8 Advanced x-ray diffractometer (Bruker, Bremen, Germany) with Cu Kα radiation (λ = 0.154 nm).

**Raman.** Raman spectra were collected by using a Lab RAMHR Evolution confocal Raman spectrometer (HORIBA, Palaiseau, France) with a 532 nm laser in the spectral of 100–4000 cm^−1^.

**XPS.** The chemical state was revealed by an AXIS Nova X-ray photoelectron spectroscopy (XPS, Shimadzu, Kyoto, Japan).

**SEM.** The morphologies of all samples were examined with an S4800 scanning electron microscope (SEM) (Hitachi, Hitachi, Japan) with an accelerating voltage of 3 kV and a Tecnai G2 f20 S-TWIN transmission electron microscope (TEM, FEI, Hillsboro, OR, USA).

**BET.** Brunauer–Emmett–Teller surface areas, pore volume, and pore size distributions of samples were measured by nitrogen adsorption analyses implemented on an ASAP 2460 BET analyzer (Micromeritics, Norcross, GA, USA).

**Electromagnetic parameters.** Electromagnetic parameters, i.e., complex permittivity (*ε*_r_ = *ε*′ − *jε*″) and complex permeability (*μ*_r_ = *μ*′ − *jμ*″), were recorded by an N5244A vector network analyzer (Agilent, Santa Clara, CA, USA) in the frequency range of 2–18 GHz. Prior to measurement, as-prepared CKF was mixed with paraffin in a mass ratio of 30:70 and then compressed into a toroidal-shaped pipe with an outer diameter of 7.00 mm, an inner diameter of 3.04 mm, and a thickness of 2.00 mm. Based on the transmission line theory, the RL of the as-fabricated samples can be determined by the following equations:(1)$$RL=20\mathrm{lg}\left|\frac{Z_{in}-Z_{0}}{Z_{in}+Z_{0}}\right|$$
(2)$$Z_{in}=Z_{0}\sqrt{\frac{\mu_{r}}{\varepsilon_{r}}}\tanh\left(j\frac{2\pi fd}{c}\sqrt{\mu_{r}\varepsilon_{r}}\right)$$
where *Z_in_* is the normalized input impedance, *Z*_0_ is the impedance of free space (377 Ω), *ε_r_* the complex permittivity, *μ_r_* is the complex permeability, *f* is the microwave frequency, *d* is the thickness of the toroidal-shaped sample, and *c* is the velocity of light.

## 3. Results

### 3.1. Overview of KF-Derived Microtube as Microwave Absorber

Possessing a typical hollow structure with a thin cell wall of 0.8~1.0 µm and large lumen of more than 80% porosity, KF is a bioproduct of an agricultural product from the *Bombaceae* family primarily grown in South Asia. Carbon microtube derived from KF has been extensively developed and utilized as an adsorbent for organic dyes, for thermal management, as sound absorption material, for catalysis, etc [13]. With natural three-dimensional tubular microstructures, the KF-derived carbon microtube may be a feasible biological candidate for the multiple scattering/reflection of penetrated electromagnetic waves [30]. Here, Kapok fibers from *Ceiba Pentandra* were chosen as the precursors for the fabrication of carbon microtube and investigating its microwave absorbing performance. As shown in Figure 1, the additive-free carbon microtube was prepared by directly carbonizing the kapok fibers in the argon atmosphere, with the designed experimental scheme of seven pyrolysis temperatures ranging from 500 °C to 800 °C. During the pyrolysis process, volatile constituents (such as CH_4_, CO_2_, and some organics) were removed, generating unique hollow microtube with numerous interior pores along the carbon skeleton [14], which is favorable for decreasing the density and boosting microwave attenuation. Among them, the carbon microtube samples with 30 wt.% loading of CKF obtained around 650 °C demonstrate the optimized MA performance, with an RL_min_ of −62.46 dB for CKF-650 and an EAB of 6.80 GHz for CKF-600, which is competitive with those of most MA materials reported previously. The morphology-associated microwave attenuation mechanisms behind this intriguing microwave absorbing performance are discussed below.

### 3.2. Nanostructures and Morphology

The crystallographic structure of the KF and CKF was characterized by XRD and Raman. As depicted in Figure 2a, KF presents a broad diffraction peak around 16.27°, which is assigned to amorphous cellulose and lignin [31]. After pyrolysis, two broad diffraction peaks at 21.95° and 33.46° corresponding to the (200) and (004) planes of the carbon phase can be observed for all CKF samples, indicating the formation of graphitic carbon. Moreover, gradually decreased strength of these characteristic peaks with the increasing of temperature indicates the increasing amount of amorphous carbon. Figure 2b shows the Raman spectra of KF and CKF obtained at different pyrolysis temperatures. While KF exhibits a flat curve with some slight fluctuations, all CKF samples demonstrate two characteristic peaks corresponding to the D band at −1326 cm^−1^ and the G band at −1588 cm^−1^, which stemmed from the lattice defects in the carbon atoms and the in-plane stretching vibration of sp^2^ carbon atoms of CKF, respectively [32]. Based on the phenomenological three-stage model proposed by Ferrari and Robertson, the absence of D-band and G-band characteristic peaks indicates the formation of nanocrystalline graphite. Generally, the ordering of carbon atoms or the degree of graphitization can be deduced from the intensity ratio of the D-band and G-band (I_D_/I_G_), and the higher the value of I_D_/I_G_ means more defects. With carbonization temperature increases, the I_D_/I_G_ value of CKF increases from 0.96 for CKF-500 to 1.07 for CKF-600, 1.17 for CKF-700, and 1.24 for CKF-800, indicating more defects and higher disordering degree [32]. Such moderately increased defects should provide not only more heterogeneous interfaces to induce dipole polarization, but also more specific charge-hopping sites to attenuate electromagnetic energy [33]. Figure 2c,d show the XPS results for KF and its carbonized counterparts. It can be seen that the XPS spectra of CKFs, similar to KF, show a significant increase in C/O atomic ratio upon increased temperature when it is compared with that of KF, implying the formation of the carbon phase. Moreover, the high-resolution C1s XPS spectra (Figure 2d) depict the strong suppression of oxygen-containing groups, including C−O (285.43 eV), C=O (286.58 eV), and O = C−O (288.53 eV), which further support the transformation of CKF [32].

The morphologies of KF and CKF under different carbonization temperatures are visualized by SEM and TEM. It can be seen from Figure 3a,a′ that KF presents a cylindrical open-ended structure with wall thicknesses of 0.45–1.26 μm and external diameters of 8.28–21.78 μm, indicating hollow volume ratios ranging from 48.93–91.91%. After pyrolysis (Figure 3b–e), all CKFs maintain their round microtubular structure with microscale channels that are much shorter than the wavelength of penetrated microwaves, which favors the reflection and scattering of electromagnetic waves [34]. When the carbonization temperature increases from 700 °C to 800 °C, there are more and more nanopores with the size of 50~500 nm in the wall (marked with white arrows in Figure 3c′–e′), which can be further verified by the TEM results in Figure 3f. Under the higher resolution (Figure 3f′), extensive sub-nanopores are noticeably observed, which together with nanopores and microtube form the hierarchical structures of CKF.

The porosities of KF and CKF were measured by N_2_ adsorption–desorption, and the relevant isotherm curves and corresponding pore size distributions are shown in Figure 4. It can be observed that all CKF samples demonstrate representative type-I isotherms, where the absorbed volume of N_2_ gradually increases with the relative press ascents. Compared with KF, CKF samples show more micropores with the increase in temperature, of which CKF-800 demonstrates the largest number of micropores. In addition, there are hysteresis loops along with upward tails in the medium-high pressure range, indicating the presence of mesopores and macropores in CKF [32]. The calculated specific surface areas of CKF-500, CKF-600, CKF-700, and CKF-800 are 4.40 m^2^/g, 5.26 m^2^/g, 6.21 m^2^/g, and 698.55 m^2^/g, respectively, which are much higher than that of KF. The increased BET area of CKF mainly results from the pyrolysis-generated enormous micropores, and higher temperatures of carbonization lead to increased numbers of multiscale pores (Figure 4b). Particularly, CK-800 possesses the largest number of micropores as well as macropores, which may be ascribed to the high-temperature-induced generation of micropores and their expansion into mesopores and even macropores, and the latter may damage or destroy the continuity of the cylindrical open-ended structure, impacting the multiple scattering and reflection at the surface [35].

### 3.3. Microwave Absorbing Performance

Generally, microwave absorbing performance is indicated by the RL, and a lower RL means a more robust microwave attenuation capacity [36]. Figure 5a–f three-dimensionally depict the RLs of CKF fabricated under different temperatures, where the matching frequency corresponding to the RL_min_ shifts gradually to the low frequency with the increase of thickness, which can be explained by the quarter wavelength matching model as shown in the following equation [2]: (3)tm=nλm/4=nc(4fm|εr||μr|12), n=1,3,5…
where |*μ_r_*| and |*ε_r_*| are the modulus of the *μ*_r_ and *ε*_r_, respectively. For a better comparison, the RLs at the optimal thickness for CKF composites were extracted and plotted in Figure 6g, and an RL below −10 dB means over 90% attenuation of the penetrated electromagnetic energy. It can be observed that the RL_min_ of CKF-550 is only −2.42 dB at the thickness of 5.5 mm and the frequency of 9.04 GHz, implying a poor microwave attenuation capacity. When the carbonization temperature increases to 650 °C, the microwave attenuation capacity of CKF is significantly enhanced, affording an RL_min_ of −47.44 dB (3.8 mm, 9.60 GHz) for CKF-600 and −62.46 dB (2.2 mm, 10.32 GHz) for CKF-650, implying 99.99 % of incident electromagnetic wave energy can be attenuated. However, as evidenced by the increased RL_min_ from −46.41 dB (2.6 mm, 9.60 GHz) for CKF-700 to −41.87 dB (2.1 mm, 9.60 GHz) for CKF-800 and −8.87 dB (1.0 mm, 16.56 GHz) for CKF-800, the microwave attenuation capacity of CKF deteriorates when the temperature is further increased, which is attributed to the destroyed microscale pore structure under excessive temperature. As an important factor that represents the frequency window for practical microwave attenuation application, the effective absorption bandwidth (EAB, RL ≤ −10 dB) of an absorber is expected to be as wide as possible [12]. Noticeably, where the EAB regions of CKF composites are enclosed by gray lines (Figure 5a–f), the EAB expands with the increase in carbonization temperatures, peaking at CKF-600 and then shrinking when the temperature further increases from 650 °C to 800 °C. Particularly, CKF-650, CKF-700, and CKF-750 demonstrate strong attenuation capacity over the broad frequency range, with EABs of 6.80 GHz (10.4–18 GHz), 4.64 GHz (10.4–18 GHz), and 4.08 GHz (10.4–18 GHz), respectively. In addition, more than 90% of the incident electromagnetic wave in the frequency ranging from 2.88 GHz to 18.00 GHz can be achieved simply by controlling the carbonization temperature of KF and/or the thickness of the CKF-based absorber. Therefore, using the same materials, the microwave absorbing performance of pure CKF absorbers is dominated by its hierarchical structure. Together with the results of SEM (Figure 3e) and BET (Figure 4), it could be deduced that the destruction of natural microscale channels and a too large average nanopore size (larger than 7 nm) is not conducive to strong, wide absorption [35]. Figure 5f summarizes typical pure-biomass-derived porous carbon and the corresponding MA capacities reported in recent literature [7,21,22,37,38,39,40,41,42], from which it can be seen that CKF-derived carbon microtube shows comparable or even superior microwave absorbing performance over a broad frequency range when it is compared with other reported absorbers.

### 3.4. Attenuation Mechanism

According to the electromagnetic energy conversion theory, the microwave absorbing performance of an absorber is primarily governed by electric loss and/or magnetic loss, which always is described by relative complex permittivity (*ε*_r_ = *ε*′ – *jε*″) and relative complex permeability (*μ*_r_ = *μ*′ – *jμ*″), where *ε*′ and *μ*′ correspond to the storage capability of electric and magnetic energies, and the *ε*″ and the *μ*″ represent the dissipation capability, respectively [43]. In addition, the dielectric and magnetic dissipation factors can be calculated using tanδ*_ε_* = *ε*″/*ε*′ and tanδ*_μ_* = *μ*″/*μ*′, respectively.

To illustrate the attenuation mechanism of CKF-derived carbon microtube, electromagnetic parameters of *ε*_r_ and *μ*_r_ are analyzed in the frequency of 2–18 GHz. As shown in Figure 6a,b, both the *ε*′ and *ε*″ values of CKF-550 and CKF-600 are nearly independent of frequency, while CKF-derived carbon microtubes obtained at 650 °C and higher exhibit a downward tendency upon frequency, which should be attributed to the enhanced polarization lagging at a higher frequency [30]. In addition, the *ε*′ and *ε*″ of CKF-700, CKF-750, and CKF-800 show several obvious resonance peaks, which are ascribed to the skin effect and are adverse for impedance matching. It is well-accepted that the presence of porous structures can decrease the effective permittivity according to the Maxwell–Garnett theory. However, CKF-800 with more carbonization-induced pore structures, as confirmed by SEM and BET results, shows the largest values of *ε*′ and *ε*″, which can be explained as follows. Generally, the electrical conductivity and polarization abilities of the material determine the *ε*′ and *ε*″ values. Initially, the carbonization endows as-prepared absorbers with interconnected carbon skeleton with numerous nanopores, which facilitates microwave attenuation via conductive loss stemming from micro-currents and capacitors. When it comes to polarization, a heterogeneous system always contains four forms including electronic, atomic, dipolar, and interfacial polarization [44]. Electronic and atomic polarization can be excluded as they occur at higher frequencies, and the dipolar polarization is also negligible as there is no evident Debye peak in the Cole−Cole plots of these four samples. Therefore, interfacial polarization dominates the polarization process, which is consistent with the previous report [45]. Carbonization, especially at a higher temperature, generates pores and defects, which act as polarization centers in the electromagnetic field, accounting for the increased *ε*′ and *ε*″ for CKF-800 [46]. Furthermore, numerous interior nanopores form numerous solid–void interfaces, where electrons would accumulate at the interfaces due to the different electrical conductivities of the solid and the void, boosting the polarization at the solid–void interfaces. The frequency dependence of the dielectric dissipation factor (tanδ*_ε_*) corresponding to the dielectric loss capability was depicted in Figure 6c, where CKF-600, CKF-650, and CKF-700 show moderate tanδ*_ε_* values around 0.37 (0.13–0.63), indicating strong dielectric loss as well as good impedance matching. However, the tanδ*_ε_* value (0.49–1.11) of CKF-800 is higher than 0.45, which could give rise to poor impedance matching characteristics and accounting for its unfavorable microwave attenuation capacity. As shown in Figure 6a′–c′, despite slight fluctuations, values of *μ*′, *μ*″, and tanδ*_μ_* are close to 1, 0, and 0, implying negligible magnetic loss [21].

Impedance matching (*Z*) and attenuation constant (*α*) are the other two vital parameters affecting the MA properties of absorbers, which indicate the capability of allowing incident electromagnetic waves to penetrate the absorber and of dissipating penetrated electromagnetic energy in other forms, respectively [43]. It is not difficult to think that a desired microwave absorber should allow as many electromagnetic waves as possible to penetrate in and then convert them into another form of energy. *Z* and *α* can be calculated as follows [6]:*Z* = |*Z_in_*/*Z*_0_|(4)
(5)$$\alpha =\frac{\sqrt{2}\pi f}{c}\sqrt{(\mu ''\varepsilon ''-\mu '\varepsilon ')+\sqrt{{\left(\mu ''\varepsilon ''-\mu '\varepsilon '\right)}^{2}+{\left(\mu '\varepsilon ''+\mu ''\varepsilon '\right)}^{2}}}$$

To further explore the microwave attenuation mechanism of CKF-derived carbon microtubes, the calculated *Z* values are presented in contour maps (Figure 7a–f). In general, the impedance matching requires a *Z* value of ~1.0 for intensive microwave attenuation. As displayed in Figure 7a, CKF-550 shows two relatively narrow strips with *Z* values from 0.8 to 1.2, delivering acceptable impedance matching performance. With the increasing temperature, CKF-600, CKF-650, and CKF-700 show improved impedance matching performance despite shrinking area (Figure 7b,c), which could stem from the moderate carbonization–induced hierarchical structure with multiscale pores [35]. However, when the temperature further increases, impedance mismatching occurs for CKF-750 and CKF-800, which can be seen in significantly reduced and even disappeared impedance matching reigns (Figure 7e,f) and ascribed to the random or coalesced pore structure. Figure 7g depicts the α values of all CKF samples, where escalating trends are observed as the frequency increases, indicating robust attenuation capacity at a higher frequency. Overall, the α values of all the CKF samples rank in the following sequence: CKF-550 < CKF-600 < CKF-650 < CKF-700 < CKF-750 < CKF-800. It should be noticed that CKF-800 possesses the highest α value over the frequency of 2–18 GHz, implying the strongest attenuation capability. However, CKF-800 displays poor microwave absorbing performance, which is ascribed to its bad impedance matching. Similarly, acceptable impedance matching but extremely low microwave dissipation leads to the poor microwave absorbing performance for CKF-550. Therefore, it is easy to conclude that balanced impedance matching and attenuation capacity account for the superior microwave absorbing performance of CKF-650. Furthermore, the relationship between RL_min_, *α* values, and *Z* values is analyzed and plotted in Figure 7g,i. Interestingly, with the Z value equal to 1, RL_min_ occurs at the right point (10.32 GHz) rather than the left point (9.04 GHz) for CKF-650 (2.2 mm), which can be further confirmed by the similar phenomenon for CKF-600, CKF-700, and CKF-750, because α value at 10.32 GHz is larger than that at 9.04 GHz [35].

As aforementioned, it can be rationally concluded that the superior microwave absorbing performance of the CKF-derived carbon microtube can stem from its hierarchical structure with nanopores, mesopores, and microtube, which affords CKF balanced impedance matching and microwave dissipation capacity. Figure 8 schematically illustrates the hierarchical-structure-associated microwave attenuation of CKF. Initially, the combination of regular microtube and interconnected carbon skeleton with numerous nanopores affords CKF improved impedance matching, allowing incident electromagnetic waves of broad frequency to penetrate the CKF-based absorber and generate multiple scattering or/and reflections. Then, the pyrolysis-induced hollow carbon skeleton endows numerous conductive pathways for electron transport and hopping, generating strong conduction loss. Moreover, interior nanopores along the carbon microtube generate numerous capacitor-like structures and solid–void interfaces, boosting the attenuation of the penetrated electromagnetic wave. Last but not the least, carbon defects and residual oxygen atoms in microtube skeletons could reinforce the dielectric loss via dipolar polarization.

## 4. Conclusions

In summary, a well-defined carbon-microtube-derived microwave absorber has been successfully prepared via a facile pyrolysis strategy by using kapok fibers as precursors, and the structure and parameters of the as-fabricated carbon microtube are effectively tailored by adjusting the carbonization temperatures. It is found that controlled carbonization affords CKF microtube with interior nanopores and mesopores, endowing it with balanced impedance matching and microwave dissipation capacity. Profiting from this, CKF demonstrates superior microwave absorbing performance, with the optimized RL_min_ of −62.46 dB (10.32 GHz, 2.2 mm) for CKF-650 and effective absorption bandwidth of 6.80 GHz (11.20–18.00 GHz, 2.8 mm) for CKF-600. Furthermore, more than 90% of the incident electromagnetic wave ranging from 2.88 GHz to 18.00 GHz can be dissipated by simply controlling the carbonization temperature of KF and/or the thickness of the carbon-microtube-based absorber. The analysis of morphology-associated electromagnetic loss behavior indicates that the enhanced microwave attenuation performance of CKF results from the hierarchical-structure-induced conductive loss, polarization, and multiple scattering/reflections.

## Figures and Tables

**Figure 1 polymers-14-04501-f001:**
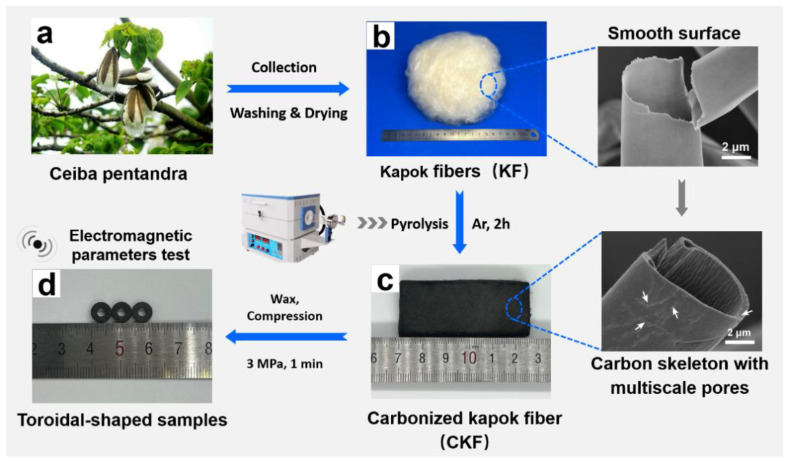
Schematic illustration of the preparation of kapok-derived carbon microtube.

**Figure 2 polymers-14-04501-f002:**
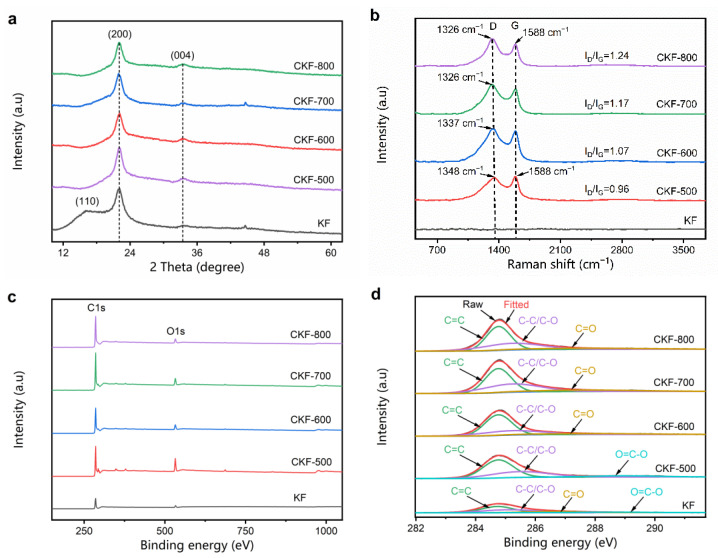
(**a**) XRD patterns, (**b**) Raman spectra, (**c**) XPS curves, and (**d**) high resolution C1s spectra.

**Figure 3 polymers-14-04501-f003:**
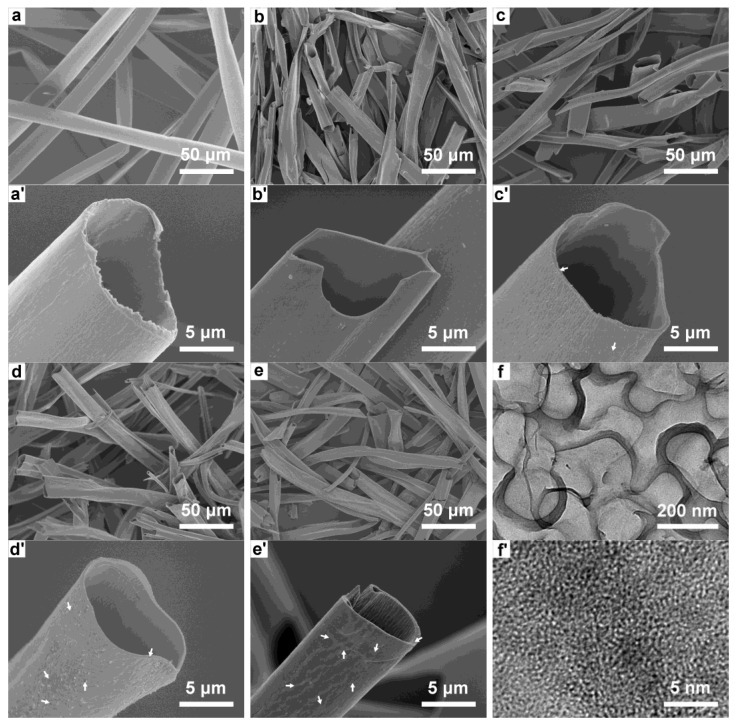
SEM images of (**a** and **a′**) KF, (**b** and **b′**) CKF-500, (**c** and **c′**) CKF-600, (**d** and **d′**) CKF-700, and (**e** and **e′**) CKF-800; (**f** and **f′**) TEM image of CKF-800.

**Figure 4 polymers-14-04501-f004:**
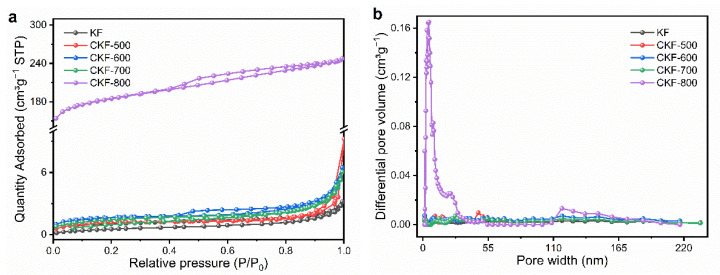
(**a**) Nitrogen adsorption–desorption isotherms and (**b**) corresponding pore size distributions of KF and CKF fabricated under different carbonization temperatures.

**Figure 5 polymers-14-04501-f005:**
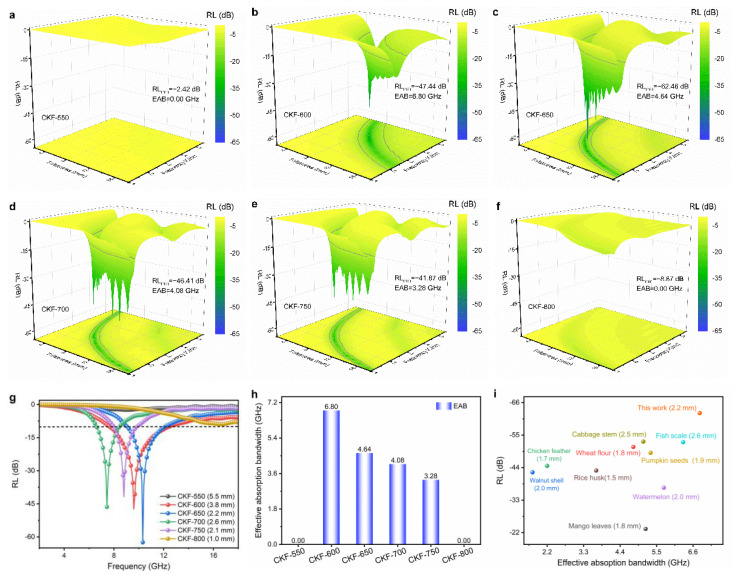
(**a**–**f**) Three-dimensional depictions of RL plots for CKF-based absorbers with different thicknesses and frequency; (**g**) RL curves and (**h**) effective absorption bandwidth of CKF at the optional thickness; (**i**) comparison of the MA capacity of reported biomass-derived carbon.

**Figure 6 polymers-14-04501-f006:**
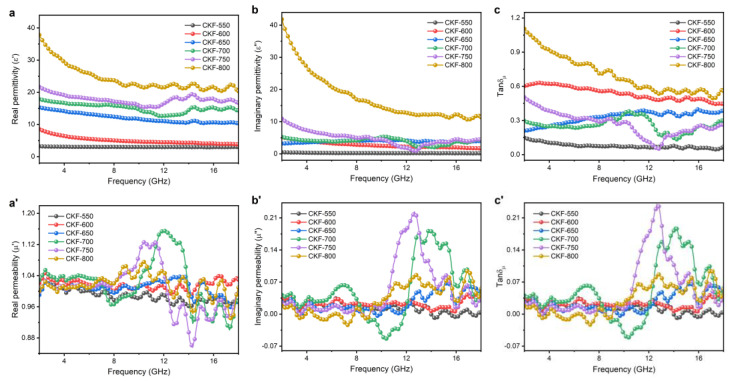
Frequency-dependent electromagnetic parameters for CKF: (**a**) real part (*ε*′) and (**b**) imaginary part (*ε*″) of complex permittivity, (**c**) dielectric loss (tan δ*_ε_*), (**a′**) the real part (*μ*′) and (**b′**) imaginary part (*μ*″) of complex permeability, and (**c′**) magnetic loss (tan δ*_μ_*).

**Figure 7 polymers-14-04501-f007:**
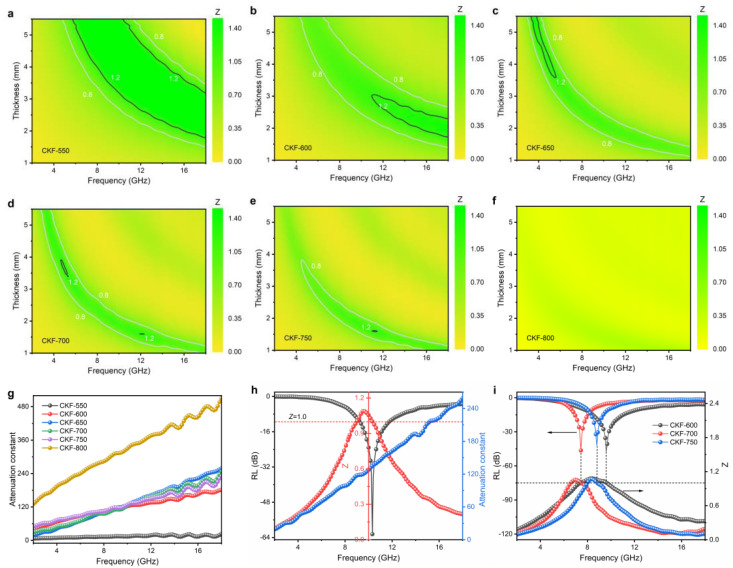
(**a**–**f**) Contour maps of *Z* values for CKF-based absorbers as a function of thickness and frequency; (**g**) calculated attenuation constant of prepared CKF samples; (**h**) the relationship between RL, attenuation constant, and Z for CKF-650; (**i**) frequency-dependent RL and Z for CKF-600, CKF 700, and CKF-750.

**Figure 8 polymers-14-04501-f008:**
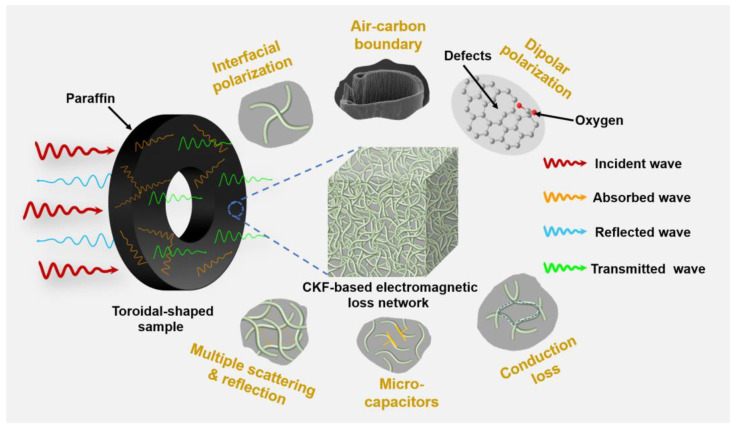
Schematic illustration of possible morphology-associated microwave attenuation for KF-derived carbon microtube.

## Data Availability

Not applicable.

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
