# Peer review of "Natural Hollow Fiber-Derived Carbon Microtube with Broadband Microwave Attenuation Capacity"

_polymers, 2022, doi:10.3390/polym14214501_

Round 1
Reviewer 1 Report
This paper “Natural Hollow Fiber-Derived Carbon Microtube with Broad-band Microwave Attenuation Capacity” by Zhao et al. represents a biomass-derived carbon material with high microwave attenuation capacity over a wide frequency range. This is an interesting and timely work, taking into account the prevailing importance of sustainable microwave absorbers in many critical and industrial applications. The results in the study look nice, and the interpretation I believe is correct. The analysis is standard for this sort of work and therefore I would recommend acceptance if the following issues were properly addressed.
1. Please further untangle the introduction based on cutting-edge bio-inspired materials used as microwave-absorbing components to fully explain why they chose KF.
2. Authors stated that the CKFs demonstrate hierarchical structures with multiscale pores. However, the reviewer cannot clearly see the structure of CKF from Figure 1 or Figure 3 in the manuscript. Authors need to replace or substitute a figure with higher magnification to support the statement.
3. What are the reasons behind the observed fluctuations even negative values in the permeability curves?
4. Please revise the grammer in manuscript carefully, and correct some typos.
Author Response
Dear Editor and Reviewer,
Thank you very much for the comments concerning our manuscript entitled “Natural Hollow Fiber-Derived Carbon Microtube with Broadband Microwave Attenuation Capacity”. These comments are all valuable and very helpful for revising and improving our paper. We have carefully checked the manuscript and made many corrections. We hope it meets the requirements for the publication of polymers. All revisions to the manuscript were blue-marked, and our point-by-point responses to the comments are as follows:
Responds to comments:
- Please further untangle the introduction based on cutting-edge bio-inspired materials used as microwave-absorbing components to fully explain why they chose KF.
Response: There are three reasons to choose KF. Initially, KF is sustainable and easily accessible, affording a green and cost-effective source of KF-based microwave absorbing materials (MAMs). Then, the presence of microscale pores in KF facilitates impedance matching via abundant air-solid interfaces and contributes to microwave attenuation via multiple reflections. Moreover, KF is easy to be processed and modified, making it versatile to develop KF-based hybrids for microwave attenuation. Please see line 58 ~line 61 and line 156 ~line 162 for more information.
- Authors stated that the CKFs demonstrate hierarchical structures with multiscale pores. However, the reviewer cannot clearly see the structure of CKF from Figure 1 or Figure 3 in the manuscript. Authors need to replace or substitute a figure with higher magnification to support the statement.
Response: Figure 1 and Figure 3 have been improved for better illustration. Specific modifications can be found from line 117, and line 207 to line 223. In addition, the original SEM images were provided for editing.
- What are the reasons behind the observed fluctuations even negative values in the permeability curves?
Response: Similar to the permittivity curves, slight fluctuations in the permeability curves may result from the resonance effect in the alternating electric field. Herein, owing to the absence of a magnetic loss component, the values of μ′ and μ″ for as-prepared CKF composites exhibit slight fluctuations around 1.0 and 0.0 (Figure 6a′ and 6b′), implying their negligible storage and dissipation capacity. Specific modifications can be found on line 335 and line 337.
- Please revise the grammar in the manuscript carefully, and correct some typos.
Response: The manuscript had been carefully checked, and special attention was paid to grammar, spelling, and sentence structures. Before resubmission, the manuscript has been refined by our professional partner from the UK.
Again, thank you very much for the comments and suggestions. If there are any problems or questions about our manuscript, please feel free to contact us.
Yours sincerely,
Pengfei Zhao
Pengfei Zhao
Associate Professor in Material Science and Engineering
Key Laboratory of Tropical Crop Products Processing of Ministry of Agriculture and Rural Affairs, Agricultural Products Processing Research Institute, Chinese Academy of Tropical Agricultural Sciences, Zhanjiang 524001, China
E-mail: pengfeizhao85ac@163.com
Tel: +86 18666729539
Reviewer 2 Report
In this study, an innovative and sustainable approach to the production of carbon microtubes with nanometer wall thickness and micrometer diameter is mentioned. The work needs to get rid of the technical report feature. I believe that the article will be published if it is revised within the framework of the general aspects stated below:
Comment-1) Please mention the advantages/disadvantages of the mechanical properties of carbon-based fibers compared to other types in the introduction part of the article and inform the readers through numerical values. Maybe it would be more informative for you to prepare a table in terms of giving visual information.
Comment-2) It's a mystery how well-known kapok fiber is, please introduce the material more.
Comment-3) Please talk extensively about microtube production methods.
Comment-4) Why not use pictures from the production stage. Actual pictures, not representative. Please attach real experiment and sample pictures to the study.
Comment-5) Does the production method you use have an international standard to which it depends?
Comment-6) The results of the study were revealed almost depending on the reflection loss RL value. Please explain more clearly and in detail the parameters given in Denkelm 1 and 2, where the RL value is calculated. There are parameters that are not explained in the formula. Indicate where and how these parameters were obtained.
Comment-7) Please focus on scientific contributions instead of repeating the experimental findings in the conclusion part.
Author Response
Dear Editor and Reviewer,
Thank you very much for the comments concerning our manuscript entitled “Natural Hollow Fiber-Derived Carbon Microtube with Broadband Microwave Attenuation Capacity”. These comments are all valuable and very helpful for revising and improving our paper. We have carefully checked the manuscript and made many corrections. We hope it meets the requirements for the publication of polymers. All revisions to the manuscript were blue-marked, and our point-by-point responses to the comments are as follows.
Responds to comments:
In this study, an innovative and sustainable approach to the production of carbon microtubes with nanometer wall thickness and micrometer diameter is mentioned. The work needs to get rid of the technical report feature. I believe that the article will be published if it is revised within the framework of the general aspects stated below:
Comment-1) Please mention the advantages/disadvantages of the mechanical properties of carbon-based fibers compared to other types in the introduction part of the article and inform the readers through numerical values. Maybe it would be more informative for you to prepare a table in terms of giving visual information.
Response: It is a good idea to take the mechanical property of kapok fiber into consideration, but this is beyond this manuscript. Due to its relatively poor mechanical property, kapok fiber is not directly used for reinforcement. Alternatively, nanocellulose from kapok fiber is a promising candidate to reinforce polymer composites, which is one of our ongoing projects. More information about the mechanical properties of carbon-based fibers will be illustrated in our next manuscript. Here, this work focus on the electromagnetic performance of biomass-derived carbon materials, and the progress of this domain is mentioned in the introduction. In addition, we also made a comparison between our work and others in the results and discussion section (Figure 5i, line 280 ~ line 284). Therefore, we do not think it is necessary to discuss the mechanical properties of carbon-based fibers, but we will add some related information later if you insist.
Comment-2) It's a mystery how well-known kapok fiber is, please introduce the material more.
Response: More information about kapok fiber is provided, please see line 156 ~ line 162.
Comment-3) Please talk extensively about microtube production methods.
Response: More information about the fabrication of microtubes was provided in the introduction (line 91 ~ line 98).
Comment-4) Why not use pictures from the production stage? Actual pictures, not representative. Please attach real experiments and sample pictures to the study.
Response: All pictures were replaced in the revised manuscript, please see Figure 1 for details (line 117). Moreover, the original SEM images were provided for editing.
Comment-5) Does the production method you use have an international standard on which it depends?
Response: To our best knowledge, there is no international standard for the fabrication of biomass-derived carbon due to the variation of biomass. However, the fabrication procedure and characterization methods used in our work are well-accepted and widely adopted in industry and academia (Polymers 2022, 14, 2014; ACS Nano 2020, 14, 595; ACS Applied Energy Materials 2019, 2, 8303; Advanced Functional Materials 2018, 28, 1803938).
Comment-6) The results of the study were revealed almost depending on the reflection loss RL value. Please explain more clearly and in detail the parameters given in Denkelm 1 and 2, where the RL value is calculated. There are parameters that are not explained in the formula. Indicate where and how these parameters were obtained.
Response: As illustrated in the section of characterization (line 144 ~ line 153), the parameters (i.e., complex permeability and complex permittivity) of toroidal-shaped samples were obtained by using an N5244A vector network analyzer. With these parameters, reflection loss (RL) at different thicknesses can be calculated based on equations (1) and (2) that are deduced from the transmission line theory, using MATLAB. For better expression, statements of the parameters of RLs are rewritten and the missed parameter (Zin) was added in the revised manuscript.
Comment-7) Please focus on scientific contributions instead of repeating the experimental findings in the conclusion part.1.
Response: Thanks for your suggestion. The conclusion was carefully modified, please see line 58 ~ line 61 and line 74 ~ line76 for further information.
Again, thank you very much for the comments and suggestions. If there are any problems or questions about our manuscript, please feel free to contact us.
Yours sincerely,
Pengfei Zhao
Pengfei Zhao
Associate Professor in Material Science and Engineering
Key Laboratory of Tropical Crop Products Processing of Ministry of Agriculture and Rural Affairs, Agricultural Products Processing Research Institute, Chinese Academy of Tropical Agricultural Sciences
E-mail: pengfeizhao85ac@163.com
Tel: +86 18666729539
Round 2
Reviewer 2 Report
The article can be published in this condition.